# Incidence and Prognostic Impact of Deleterious Germline Mutations in Primary Advanced Ovarian Carcinoma Patients

**DOI:** 10.3390/cancers15092534

**Published:** 2023-04-28

**Authors:** Majdi Imterat, Philipp Harter, Kerstin Rhiem, Florian Heitz, Stephanie Schneider, Nicole Concin, Malak Moubarak, Julia Welz, Vasileios Vrentas, Alexander Traut, Eric Hahnen, Rita Schmutzler, Andreas du Bois, Beyhan Ataseven

**Affiliations:** 1Department of Gynaecology and Gynaecologic Oncology, Kliniken Essen Mitte (KEM), 45136 Essen, Germany; p.harter@gmx.de (P.H.); f.heitz@kem-med.com (F.H.); st.schneider@kem-med.com (S.S.); nicole.concin@i-med.ac.at (N.C.); m.moubarak@kem-med.com (M.M.); j.welz@kem-med.com (J.W.); v.vrentas@kem-med.com (V.V.); a.traut@kem-med.com (A.T.); prof.dubois@googlemail.com (A.d.B.); ataseven@gmx.net (B.A.); 2Department of Gynaecologic Oncology, Hadassah Medical Centers, Faculty of Medicine, Hebrew University of Jerusalem, Kalman Ya’Akov Man St., Jerusalem 91905, Israel; 3Center for Familial Breast and Ovarian Cancer, Center for Integrated Oncology (CIO), Medical Faculty, University Hospital Cologne, 50931 Cologne, Germany; kerstin.rhiem@uk-koeln.de (K.R.); eric.hahnen@uk-koeln.de (E.H.); rita.schmutzler@uk-koeln.de (R.S.); 4Department for Gynecology with the Center for Oncologic Surgery Charité Campus Virchow-Klinikum, Charité—Universitätsmedizin Berlin, Corporate Member of Freie Universität Berlin, Humboldt-Universität zu Berlin, and Berlin Institute of Health, 10117 Berlin, Germany; 5Academic Department of Gynecology, Gynecologic Oncology and Obstetrics, Klinikum Lippe, Medical School, University Medical Center East Westphalia-Lippe, Bielefeld University, 33615 Bielefeld, Germany

**Keywords:** *BRCA1/2*, *RAD51C/D*, *BRIP1*, *PALB2*, ovarian cancer, survival

## Abstract

**Simple Summary:**

The availability of multiple gene panel testing allows us to identify germline pathogenic variants of validated cancer genes. We evaluated the prevalence and clinical/prognostic impact of deleterious germline mutations in OC patients. Germline panel testing should be performed for all patients with ovarian cancer. Better prognosis was found for germline mutated ovarian cancer patients. Endometrioid and clear cell histology subtypes show more deleterious mutations in other genes.

**Abstract:**

Data on deleterious variants in genes other than *BRCA1/2* remain limited. A retrospective cohort study was performed, including primary OC cases with TruRisk^®^ germline gene panel testing between 2011 and 2020. Patients with testing after relapse were excluded. The cohort was divided into three groups: (A) no mutations, (B) deleterious *BRCA1/2* mutations, and (C) deleterious mutations in other genes. A total of 702 patients met the inclusion criteria. Of these 17.4% (n = 122) showed *BRCA1/2* mutations and a further 6.0% (n = 42) in other genes. Three-year overall survival (OS) of the entire cohort was significantly longer in patients with germline mutations (85%/82.8% for cohort B/C vs. 70.2% for cohort A, *p* < 0.001) and 3-year progression-free survival (PFS) only for cohort B (58.1% vs. 36.9%/41.6% in cohort A/C, *p* = 0.002). In multivariate analysis for the subgroup of advanced-stages of high-grade serous OC, both cohorts B/C were found to be independent factors for significantly better outcome, cohort C for OS (HR 0.46; 95% CI 0.25–0.84), and cohort B for both OS and PFS (HR 0.40; 95% CI 0.27–0.61 and HR 0.49; 95% CI 0.37–0.66, respectively). Germline mutations were detected in a quarter of OC patients, and a quarter of those in genes other than *BRCA1/2*. Germline mutations demonstrate in our cohort a prognostic factor and predict better prognosis for OC patients.

## 1. Introduction

Ovarian carcinoma (OC) is one of the most aggressive women’s cancers, with decreasing incidence and mortality rate over the past decade [1]. In 2020, the Global Cancer Observatory reported a worldwide incidence of 313.959 (6.6 per 100,000) new cases and a total of 207,252 deaths (4.2 per 100,000) [2]. The epidemiology of this cancer shows relevant differences between ethnicities and countries due to multiple factors, including genetic and environmental [3]. Unfortunately, detection and primary diagnosis occur mostly in advanced-stages due to nonspecific symptoms and lack of sufficient predictive screening modalities.

The prevalence of genetic predisposition and hereditary mutations for cancers are various among populations. The identification of high-risk gene carriers allows crucial decisions for focused monitoring. This leads to early detection and allows interventions for prevention such as risk-reducing surgeries [4,5].

Germline mutations have been reported in up to a quarter of OC cases [6,7]. Furthermore, knowledge about deleterious mutations has been recognized as mandatory and essential for therapeutic and prognostic implications [8]. However, the heterogeneous indication for germline testing may depend on the national prevalence of deleterious *BRCA1/2* mutations in each population [6]. The European Society of Gynaecological Oncology (ESGO) and the European Society of Medical Oncology (ESMO) guidelines suggest *BRCA1/2* testing for all OC patients, except those with mucinous histology [8]. The Society of Gynecologic Oncology (SGO) recommends generally *BRCA1/2* testing [9]. On the other hand, the American Society for Clinical Oncology (ASCO) recommends germline multiple gene testing of further other clinically validated mutations for all OC patients [10].

In the last decade, the role of targeted maintenance therapy became a main issue, and several phase III studies have shown a prognostic benefit of poly (ADP-ribose) polymerase inhibitors (PARP-i) in selected cohorts with genomic mutations/instability (germline or somatic) [11,12,13]. Thus, multiple gene panel testing has been performed more often to identify genomic and molecular alterations, which could be optional disease drivers and markers for benefit from targeted maintenance therapies.

Germline *BRCA1/2* mutations, the most common mutation in OC patients, demonstrate a well-known prognostic factor, likely because of the good response to platin-based chemotherapy and maintenance therapy with PARP-i, which nowadays is approved for this patient group. A previous study showed a significantly better outcome (overall survival (OS) and progression-free survival (PFS)) for germline *BRCA1/2*-mutated OC patients in comparison with non-mutated patients [14]. Moreover, gene panel testing enables an extended examination of numerous high-risk hereditary mutations, such as *RAD51C/D, BRIP1, PALB2*, and *MSH6* [4,15,16,17]. Deleterious mutations of genes other than *BRCA1/2* were reported in 4–6% of OC patients. However, the role and impact of those mutations on the prognosis remains unclear [7,15].

The purpose of this current study was to evaluate the prevalence and clinical impact of deleterious germline mutations, in *BRCA1/2* and other genes, on survival of OC patients.

## 2. Methods

We conducted a retrospective cohort study of patients with primary diagnosis of epithelial OC in any stage who underwent treatment in our tertiary gynecologic oncology center (Kliniken Essen-Mitte, Essen, Germany) between 2011 and 2020. The therapy strategy consisted of primary debulking surgery if feasible, 6 cycles of platinum-based chemotherapy (usually carboplatin AUC 5 and paclitaxel 175 mg/m^2^, every 3 weeks), and maintenance therapy (in stage FIGO III/IV) with bevacizumab (if no contraindication), PARP-I, or both. The surgical strategy aimed at macroscopic complete resection was performed by a specialized team including at least one board-certified gynecologic oncologist.

Patient information, demographic/clinical, and tumor-related characteristics were extracted from medical records and retrieved from our prospectively maintained clinical tumor registry. Only patients with germline gene panel testing were included in the analysis. All patients were tested at primary diagnosis. Patients with testing after relapse were excluded.

All patients provided signed informed consent prior for documentation of clinical data in the tumor registry for clinical research, quality assurance analyses, and publication. In addition, all patients gave written informed consent according to the German genetic diagnostics law after detailed counseling. Genetic testing was performed according to German guidelines [7] and as gene panel analyses in cooperation with the German Consortium for Hereditary Breast and Ovarian Cancer (GC-HBOC) in Cologne. Genetic testing and variant classification (at GC-HBOC) was performed as described previously [6].

From 2015 onwards, genetic analysis within the GC-HBOC was based on the TruRisk^®^ gene panel test [18], including additionally other clinically relevant genes listed in the National Comprehensive Cancer Network (NCCN) breast and ovarian genetic/familiar cancer guidelines: *RAD51C*, *RAD51D*, *PALB2*, *BRIP1*, *CHEK2*, *MLH1*, *MSH2*, *MSH6*, *PMS2*, *TP53*, *ATM*, *CDH1*, *NBN*, *EPCAM*, *PTEN*, *STK11,* and *BARD1* [19]. Indication of genetic counseling and testing were previously based on strict orientated individual and family history for breast/ovarian cancer. However, counseling and testing has been offered since 2017 for every OC patient based on the inclusion criteria of the GC-HBOC [7].

Based on the results of the gene panel testing, our cohort was divided into three groups: (A) no deleterious germline mutation, (B) *BRCA1/2* mutation, and (C) other deleterious germline mutation.

## 3. Statistical Analysis

Statistical analysis was performed using GraphPad Prism (version 7) and SPSS (version 27.0, IBM Corporation, New York, NY, USA) software. Initial analysis compared background, and tumor characteristics between the different study groups, using the chi squared test for categorical data or based on variable characteristics and normal distribution *t*-test or non-parametric Mann–Whitney test, and ANOVA or Kruskal–Wallis H for continuous variables.

Patients’ medical history and tumor characteristics were analyzed and compared for the entire cohort and as a sensitivity analysis for subgroup advanced-stages: International Federation of Gynecology and Obstetrics (FIGO) III and IV and high-grade serous OC. Kaplan–Meier survival curves were constructed for this subgroup cohort—OS (defined as the duration of patient survival from the date of primary diagnosis) and PFS (defined as the time from primary diagnosis date until first disease recurrence or patient death without recurrence)—and compared using the Cox–Mantel log-rank test. In addition, uni- and multivariate Cox regression analyses for OS and PFS separately using the Wald test were performed for the latter patient subgroup of advanced-stage high-grade serous histology, with and without testing after recurrence, after adjusting for relevant confounders. Follow-up was performed according to the German guideline’s recommendation.

All analyses were regarded as hypothesis-generating, and a *p*-value of 0.05 was interpreted as being significant.

## 4. Results

Entire cohort

Patient characteristics

During the study period, 1584 patients with primary diagnosed OC were treated in our center. Of these, 702 patients (44.3%) fulfilled the inclusion criteria for this analysis with evaluable germline gene panel testing results. A majority (94%, n = 660) had been tested within 2 years from the primary diagnosis.

Table 1 presents baseline patients’ demographic and tumor characteristics in the different groups of the entire cohort: 76.6% (n = 538) had no deleterious mutation (cohort A); 17.4% (n = 122) a deleterious *BRCA1/2* mutation (cohort B): *BRCA1*- 69.7% (n = 85), *BRCA2*- 30.3% (n = 37); and 6% (n = 42) showed another deleterious mutation (cohort C). The following gene mutations were detected in cohort C: 19.0% (n = 8) *RAD51C*, 14.3% (n = 6) *MSH6*, 14.3% (n = 6) *BRIP1*, 11.9% (n = 5) *RAD51D*, 11.9% (n = 5) *PALB2*, 9.5% (n = 4) *CHEK2*, 4.8% (n = 2) *ATM*, 4.8% (n = 2) *PMS2*, 4.8% (n = 2) *TP53*, 2.4% (n = 1) *MLH1*, and 2.4% (n = 1) *BARD1*.

The median patient age was younger in cohort B (57 years) than in other cohorts (cohort A 60 years and cohort C 63 years, *p* = 0.005). Rate of high age-adjusted Charlson Comorbidity Index (ACCI ≥ 4) was significantly lower in cohort B (4.9%) than in other cohorts (14.9% and 21.4% in cohort A and C, *p* = 0.013). The rate of previous malignancy, mainly breast cancer, was significantly higher in patients with germline mutation (21.3%, n = 26 in cohort B and 33.4%, n = 14 in cohort C vs. 13%, n = 70 in cohort A, *p* = 0.001). Most cases consisted of high-grade serous histology (80.3%, n = 565), which was significantly more common in cohort B (97.5%, n = 119 vs. 73.8%, n = 31 in cohort C and 77%, n = 414 in cohort A, *p* < 0.001). Only one patient (1.5%) with low-grade OC histology had *BRCA1/2* mutation, and a further five patients (7.6%) in other genes. Mucinous OC patients showed no germline mutations (n = 14, 2%). No significant differences between the study groups were detected for performance status, FIGO stage, surgery timing, or residual disease after debulking surgery.

In terms of systemic therapy, no significant differences were found in the application of first-line chemotherapy (a majority received combination chemotherapy with carboplatin and paclitaxel: 85.3%, n = 599) and bevacizumab maintenance therapy (59.5%, n = 418).

However, the use of PARP-i maintenance therapy in the first-line setting was significantly higher in patients with *BRCA1/2* mutation (cohort B 32%, n = 39 vs 13.3%, n = 6 in cohort C and 9.9%, n = 53 in cohort A, *p* < 0.001).

Survival outcome

Median OS/PFS was 57.5/26.0 months for cohort A, (not reached)/48.3 months for cohort B, and 70.7/26.3 for cohort C, which was significantly different between the groups: *p* < 0.001/*p* = 0.002, respectively.

2.Advanced-stage high-grade serous cohort

Patient characteristics

Due to very heterogeneous survival outcomes of different OC stages/histology, we conducted separate analyses for the subgroup of advanced-stage (III–IV), high-grade serous histology (n = 513): 73.3% (n = 376) had no deleterious mutation (cohort A); 20.9% (n = 107) a deleterious *BRCA1/2* mutation (cohort B); and 5.8% (n = 30) showed another deleterious mutation (cohort C).

Patient characteristics did not substantially differ between the study groups. However, the incidence of complete tumor resection was lower in cohort C (55.6%, n = 15) than other cohorts (cohort A 71.5% and cohort B 79%, Table 2). In addition, the rate of PARP-i maintenance therapy in the first-line setting was significantly higher in patients with germline mutation (cohort B 34.6%, n = 37 vs 20%, n = 6 in cohort C and 13%, n = 49 in cohort A, *p* < 0.001), as well after the first relapse (cohort B 49%, n = 25 vs 55%, n = 11 in cohort C and 28.7%, n = 62 in cohort A, *p* = 0.003, Table 2).

Survival outcome

The Kaplan–Meier curve (Figure 1) showed significantly longer median OS for both B and C cohorts (91.1 and 70.7 months, respectively, vs 45.7 in cohort A, log rank < 0.001). Superior median PFS (Figure 2) could be demonstrated only for cohort B (37.6 months vs 23.5 for cohort C, and 22.4 in cohort A, log rank < 0.001).

Multivariate analysis for survival

In multivariate Cox regression analysis (Table 3) for this subgroup, both cohorts B and C (vs cohort A as reference) were found to be independent factors for a significantly superior outcome: cohort C for OS (hazard ratio [HR] 0.46; 95% CI 0.25–0.84, *p* = 0.011), and cohort B for both OS and PFS (HR 0.40; 95% CI 0.27–0.61, *p* < 0.001; and HR 0.49; 95% CI 0.37–0.66, *p* < 0.001, respectively), while controlling for previous malignancy, ACCI, FIGO stage, surgery timing and residual tumor.

Further multivariate analysis for this subgroup after excluding patients that had been tested after diagnosis of recurrence showed similar findings (Table 4).

## 5. Discussion

Deleterious germline mutations were found in almost a quarter (23.4%) of the entire cohort included in the current study. In more than a quarter of these (25.6%) a mutation in genes other than *BRCA1/2* could be identified. In our cohort of patients with advanced-stage high-grade serous OC and germline gene panel testing, deleterious mutations proved to be significantly associated with a favorable prognosis: *BRCA1/2* for both OS and PFS and other genes for OS.

Survival impact of germline deleterious mutations, particularly *BRCA1/2*, in OC patients showed conflicting results. While some authors have shown a superior prognosis for germline *BRCA1/2*-mutated patients [20,21,22], others could not prove an association regarding long-term survival [23,24]. A multicenter case–control study of the Australian Ovarian Cancer Study Group found that germline *BRCA1/2* mutation was an independent predictor of improved OS and PFS [25]. Pennington et al. demonstrated a similar incidence and proportion of deleterious germline mutations (26% in other genes), with better OS for both mutated patient groups—*BRCA1/2* and other genes [26]. They predicted germline mutations were a significant factor for better first-line platinum-based chemotherapy response, the main hypothesis for the favorable outcome in these subgroups. Our finding of longer OS for deleterious mutation in other genes may also be addressed by adjuvant chemo- and maintenance therapy.

Following the results of the SOLO-1 study published 2018 [11], PARP-i (olaparib) was approved as primary maintenance therapy for *BRCA1/2*-mutated patients with advanced-stage high-grade OC. This fact supports the higher rate of PARP-i application for our *BRCA1/2*-mutated cohort as first-line treatment, which may be a further hypothetic explanation for the superior short-term outcome (PFS) in this subgroup.

Furthermore, recently published studies confirmed the activity of PARP-i also in breast cancer patients with other germline mutated genes such as *PALB2* [27,28]. Other trials showed survival benefit from PARP-i for *ATM*-mutated patients with different malignancies [29,30]. The exact underlying mechanism is still unclear; however, genome instability and diminished DNA-repair pathways can increase sensitivity to PARP-i therapy. If similar effects can be expected for OC patients, further trials focusing on maintenance treatment and deleterious germline mutations other than *BRCA1/2* are necessary.

Recent ESMO guidelines for risk reduction and screening of individuals with a hereditary breast and ovarian cancer syndrome suggested a broad classification of other genes according to their risk for OC [5]: high-risk genes such as *RAD51C/D*, moderate-risk genes such as *PALB2*, and no-risk mutations such as *CHEK2* and *TP53*. A majority of deleterious other genes mutations of the present study were high-risk genes for OC.

Considering histology-related mutation rate, in agreement with our findings, high-grade serous histology stratifies the highest incidence of germline deleterious mutations, particularly in *BRCA1/2* genes, while other histology shows a very low rate [31]. In our cohort, mucinous histology has never shown germline mutations, supporting the abovementioned ESGO/ESMO germline testing recommendation. The proportion of endometrioid and clear-cell histology subtypes was higher in other gene deleterious mutations, related to the frequent Lynch syndrome gene mutation [15,31,32], predominantly in *MSH6*.

### Strengths and Limitations

The large number of participants (n = 702) in a non-selected cohort with germline gene panel testing in primary diagnosed EOC contributes to the strength of this study. In addition, the continuous follow-up allowed close evaluation of any events that occurred. Nevertheless, the study has several notable limitations, mainly due to its retrospective design. Thus, the results suggest an association only, rather than causation or underlying pathogenesis. Several tests have been performed after recurrence. Additionally, we focused on germline mutation in the mentioned genes, given the nature of our database and missed the data on somatic mutation or homologous recombination deficiency. Due to the scarcity of mutations in some genes, the small samples in the different subgroups limit the interpretation of our findings.

## 6. Conclusions

To the best of our knowledge, this is the largest recent single-center study to evaluate the outcome impacts of deleterious germline mutations on primary OC compared to non-mutated patients. We conclude that genetic counseling and germline panel testing provide essential information for affected OC patients and their families. Hereditary deleterious germline mutations in both *BRCA1/2* or other genes were identified in a quarter of OC patients, and our data suggest a favorable prognosis in advanced-stage high-grade serous OC. Germline panel testing use to be focused on specific genes depending on histology subtype, particularly on genes other than *BRCA1/2* for endometrioid and clear-cell cases. As the number of patients with mucinous OC was limited in our cohort, we could not draw any firm conclusions for this subgroup. Finally, the effect of targeted maintenance therapies for other germline-mutated OC patients necessitates further investigation.

## Figures and Tables

**Figure 1 cancers-15-02534-f001:**
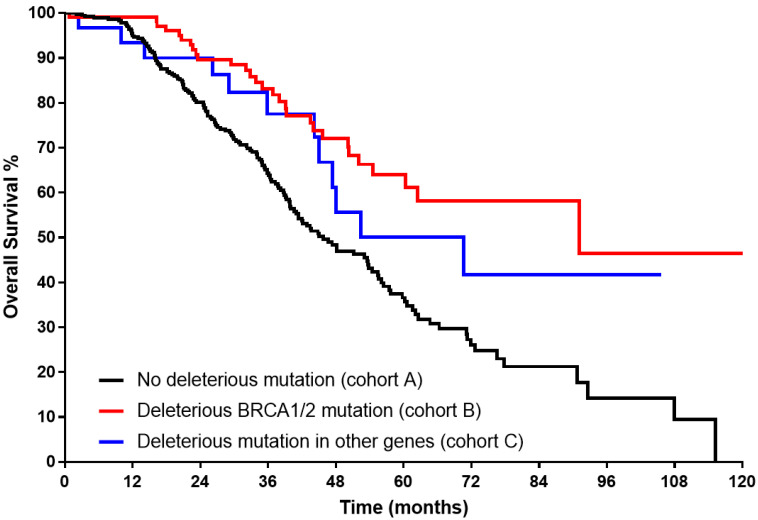
Kaplan–Meier curve for overall survival of each study group in advanced-stage high-grade serous histology (log rank < 0.001).

**Figure 2 cancers-15-02534-f002:**
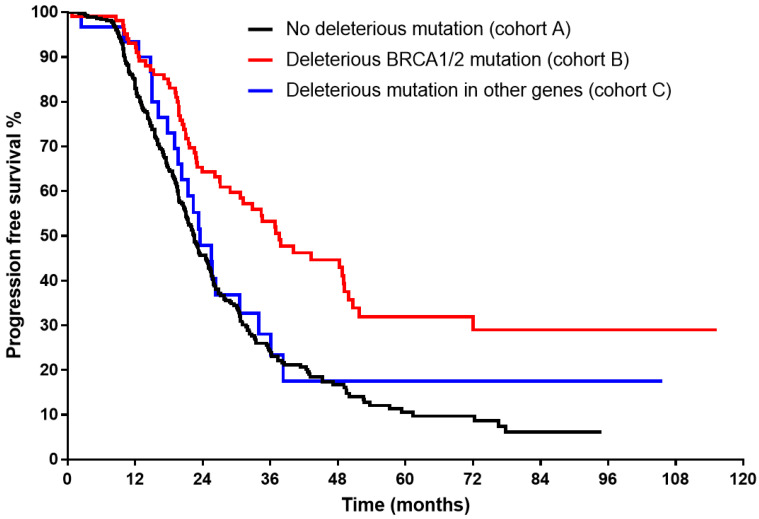
Kaplan–Meier curve for progression-free survival of each study group in advanced-stage high-grade serous histology (log rank < 0.001).

**Table 1 cancers-15-02534-t001:** Demographic and tumor characteristics of the entire cohort in the different study groups.

Characteristic	Total 100% (n = 702)	No Deleterious Mutation 76.6% (n = 538) Cohort A	Deleterious *BRCA 1/2* Mutation 17.4% (n = 122) Cohort B	Other Deleterious Mutation 6.0% (n = 42) Cohort C	*p*-Value
Patient age, Median (range)	59 (18–87)	60 (18–87)	57 (38–85)	63 (30–85)	0.011
Test Interval (Years):					0.553
≤2	94.0 (660)	94.1 (506)	95.1 (116)	90.5 (38)	
>2	6.00 (42)	5.90 (32)	4.90 (06)	9.50 (04)	
Performance status (ECOG *):					0.089
0	93.7 (658)	93.1 (501)	97.5 (119)	90.5 (38)	
>0	6.30 (44)	6.90 (37)	2.50 (03)	9.50 (04)	
ACCI **:					0.013
0–1	44.2 (310)	43.1 (232)	52.5 (64)	33.3 (14)	
2–3	42.3 (297)	42.0 (226)	42.6 (52)	45.2 (19)	
>4	13.5 (95)	14.9 (80)	4.90 (06)	21.4 (09)	
History of previous malignancy:					0.001
Breast cancer	8.7 (61)	6.5 (35)	15.6 (19)	16.7 (7)	
Others	7.0 (49)	6.5 (35)	5.7 (7.0)	16.7 (7)	
Surgery type:					0.225
Primary debulking	74.5 (523)	75.3 (408)	71.3 (87)	66.7 (28)	
Interval debulking	22.6 (159)	21.4 (115)	27.0 (33)	26.2 (11)	
Diagnostic	2.80 (20)	2.80 (15)	1.60 (02)	7.10 (03)	
Histology subgroups:					<0.001
High-grade serous (G2–3)	80.3 (564)	77.0 (414)	97.5 (119)	73.8 (31)	
Low-grade serous (G1)	5.8 (41)	7.2 (39)	0.8 (01)	2.4 (01)	
High-grade endometroid (G3)	2.4 (17)	2.4 (12)	0.8 (01)	7.1 (03)	
Low-grade endometroid (G1–2)	3.6 (25)	3.9 (21)	0	9.5 (04)	
Mucinous (G1–3)	2.0 (14)	2.6 (14)	0	0	
Clear cell	4.0 (28)	4.6 (25)	0	7.1 (03)	
MMMT ***	1.7 (12)	2.0 (11)	0.8 (01)	0	
FIGO stage:					0.063
I	9.0 (63)	9.9 (53)	4.1 (5)	11.9 (5)	
II	7.1 (50)	7.4 (40)	5.7 (7)	7.1 (3)	
III	37.7 (265)	35.7 (192)	41.8 (51)	52.4 (22)	
IV	46.2 (324)	47 (253)	48.4 (59)	28.6 (12)	
Residual disease after:					0.365 0.941
- Primary debulking	80.1 (419)	79.9 (326)	85.1 (74)	67.9 (19)	
No	14.3 (75)	14.5 (59)	11.5 (10)	21.4 (06)	
1–9 mm	5.50 (29)	5.60 (23)	3.40 (03)	10.70 (03)	
>10 mm					
- Interval debulking					
No	73.0 (116)	73.0 (84)	72.7 (24)	72.7 (08)	
1–9 mm	22.0 (35)	21.7 (25)	21.2 (07)	27.3 (03)	
>10 mm	5.0 (08)	5.2 (06)	6.1 (02)	0	
First-line chemotherapy:					0.162
Single-agent carboplatin	9.4 (66)	10.6 (57)	4.1 (5)	9.5 (4)	
Carboplatin + paclitaxel	85.3 (599)	83.8 (451)	92.6 (113)	83.3 (35)	
Others	1.7 (12)	1.5 (08)	2.5 (03)	2.4 (01)	
None	3.6 (25)	4.1 (22)	0.8 (01)	4.8 (02)	
First-line maintenance therapy:					
Bevacizumab	59.5 (418)	58.7 (316)	63.1 (77)	59.5 (25)	0.673
PARP-I ****	14.4 (101)	10.0 (54)	33.6 (41)	14.3 (6)	<0.001
Both	8.30 (58)	6.30 (33)	18.0 (22)	7.10 (03)	<0.001
PARP-i/Placebo	3.1 (22)	3.0 (16)	3.3 (4)	4.8 (2)	0.810
Follow-up time, median (months)					43
Recurrence					57.1 (401)
Overall survival:					<0.001
Median (months)	62.1	57.5	N.R.	70.7	
3-year (%)	73.4	70.2	85.0	82.8	
Progression-free survival:					0.002
Median (months)	28.1	26.0	48.3	26.3	
3-year (%)	41.1	36.9	58.1	41.6	

Data are presented as % (n) or median (range); significance was measured using chi squared. N.R.: not reached, * Eastern Cooperative Oncology Group, ** age-adjusted Charlson Comorbidity Index, *** carcinosarcoma and malignant mixed mesodermal tumor of the ovaries, **** poly(ADP-ribose) polymerase inhibitors.

**Table 2 cancers-15-02534-t002:** Selected primary and recurrence disease characteristics of the subgroup advanced-stage high-grade serous OC.

Characteristic	Total 100% (n = 513)	No Deleterious Mutation 73.3% (n = 376) Cohort A	Deleterious *BRCA 1/2* Mutation 20.9% (n = 107) Cohort B	Other Deleterious Mutation 5.8% (n = 30) Cohort C	*p*-Value
Test Interval (Years):					0.553
≤2	93.6 (480)	93.9 (353)	94.4 (101)	86.7 (26)	
>2	6.40 (33)	6.10 (23)	5.60 (06)	13.30 (04)	
Primary diagnosis:					0.346
Primary debulking	67.8 (348)	68.4 (257)	68.2 (73)	60.0 (18)	
Interval debulking	28.3 (145)	27.6 (104)	29.9 (32)	30.0 (9)	
Diagnostic surgery	3.9 (20)	4.0 (15)	1.9 (2)	10.0 (3)	
Residual disease after:					0.074 0.913
- Primary debulking					
No	72.7 (253)	72.7 (184)	82.2 (60)	50.0 (09)	
1–9 mm	19.8 (69)	20.6 (53)	13.7 (10)	33.3 (06)	
>10 mm	7.5 (26)	7.8 (20)	4.1 (03)	16.7 (03)	
- Interval debulking					
No	71.0 (103)	71.2 (74)	71.9 (23)	66.7 (06)	
1–9 mm	24.1 (35)	24.0 (25)	21.9 (07)	33.3 (03)	
>10 mm	4.8 (07)	4.8 (05)	6.3 (02)	0	
First-line chemotherapy:					0.785
Single-agent carboplatin	3.5 (18)	3.7 (14)	1.9 (2)	6.7 (2)	
Carboplatin + paclitaxel	93.4 (479)	93.4 (351)	94.4 (101)	90.0 (27)	
Others	1.9 (10)	1.6 (6)	2.8 (3)	3.3 (1)	
None	1.2 (6)	1.3 (5)	0.9 (1)	0	
First-line maintenance therapy:					
Bevacizumab	71.2 (365)	71.3 (268)	71.0 (76)	70.0 (21)	0.989
PARP-i *	18.7 (96)	13.6 (51)	36.4 (39)	20.0 (6)	<0.001
Both	11.1(57)	8.50 (32)	20.6 (22)	10.0 (03)	0.002
PARP-i/Placebo	4.1 (21)	4.0 (15)	3.7 (4)	6.7 (2)	0.759
First Recurrence:					60.2 (309)
Secondary debulking surgery					19.5 (56)
Residual disease after debulking:					0.418
No	72.7 (32)	71.4 (20)	66.7 (8)	100 (4)	
Yes	27.3 (12)	28.6 (8)	33.3 (4)	0	
Second line chemotherapy:					0.081
Platinum-based mono	6.2 (16)	6.2 (12)	4.3 (2)	10.0 (2)	
Platinum-based combination	78.8 (204)	75.1 (145)	93.5 (43)	80.0 (16)	
Others	7.7 (20)	10.4 (20)	0	0	
None	7.3 (19)	8.3 (16)	2.2 (1)	10.0 (2)	
Second line maintenance therapy:					
Bevacizumab	13.6 (36)	13.6 (27)	13.0 (6)	15.0 (3)	0.978
PARP-i	34.2 (98)	28.7 (62)	49.0 (25)	55.0 (11)	0.003
Follow-up time, median (months)					43.1
Overall survival:					<0.001
Median (months)	53.5	45.7	91.1	70.7	
3-year (%)	67.3	64.3	83.1	77.5	
Progression-free survival:					<0.001
Median (months)	23.5	22.4	37.6	23.5	
3-year (%)	30.9	24.3	53.2	28.1	

Data are presented as % (n) or median; Significance was measured using chi squared. Missed cases were excluded. * Poly(ADP-ribose) polymerase inhibitors.

**Table 3 cancers-15-02534-t003:** Uni- and multivariate Cox analysis for progression-free and overall survival of the entire advanced-stage III–IV, high-grade serous histology patients.

Covariates	Patients, n	Progression-Free Survival				Overall Survival			
		Univariate HR (95% CI)	*p*-Value (Wald Test)	Multivariate HR (95% CI)	*p*-Value (Wald Test)	Univariate HR (95% CI)	*p*-Value (Wald Test)	Multivariate HR (95% CI)	*p*-Value (Wald Test)
Gene panel test result:									
No mutations	376	Ref.		Ref.		Ref.		Ref.	
BRCA 1/2 mutation	107	0.49 (0.37–0.65)	<0.001	0.49 (0.37–0.66)	<0.001	0.41 (0.27–0.60)	<0.001	0.40 (0.27–0.61)	<0.001
Other mutations	30	0.83 (0.54–1.29)	0.412	0.71 (0.45–1.09)	0.119	0.61 (0.34–1.10)	0.098	0.46 (0.25–0.84)	0.011
History of malignancy:									
No	439	Ref.		Ref.		Ref.		Ref.	
Breast cancerothers	42	0.81 (0.54–1.21)	0.297	0.73 (0.48–1.11)	0.147	1.03 (0.62–1.73)	0.900	1.26 (0.74–2.16)	0.387
	32	1.78 (1.20–2.63)	0.004	1.83 (1.24–2.72)	0.003	1.89 (1.17–3.04)	0.009	2.00 (1.24–3.23)	0.004
Surgery timing and residual disease (RD):									
Primary debulking without RD	253	Ref.		Ref.		Ref.		Ref.	
Primary debulking with RD	95	2.85 (2.14–3.79)	<0.001	2.90 (2.17–3.86)	<0.001	2.79 (1.97–3.96)	<0.001	2.89 (2.03–4.12)	<0.001
Interval debulking without RD	103	1.87 (1.40–2.48)	<0.001	1.86 (1.40–2.48)	<0.001	1.83 (1.24–2.70)	0.003	1.80 (1.22–2.66)	0.003
Interval debulking with RD	42	4.15 (2.86–6.03)	<0.001	4.49 (3.08–6.53)	<0.001	5.82 (3.72–9.11)	<0.001	6.29 (4.10–9.88)	<0.001
Diagnostic surgery	20	5.47 (3.28–9.13)	<0.001	6.46 (3.81–10.9)	<0.001	3.66 (1.94–6.93)	<0.001	3.77 (1.77–7.21)	<0.001
FIGO stage:									
III	219	Ref.		Ref.		Ref.		Ref.	
IV	294	1.41 (1.13–1.76)	0.002	1.18 (0.94–1.47)	0.156	1.46 (1.10–1.94)	0.009	1.20 (0.90–1.61)	0.219
ACCI * score:									
0–1	198	Ref.		Ref.		Ref.		Ref.	
2–3	240	1.01 (0.80–1.28)	0.912	0.97 (0.77–1.23)	0.832	1.23 (0.91–1.66)	0.186	1.18 (0.86–1.60)	0.306
>4	75	1.52 (1.12–2.08)	0.008	1.16 (0.82–1.63)	0.412	1.93 (1.30–2.87)	0.001	1.47 (0.93–2.33)	0.101

Data are presented as hazard ratio (HR) (95% CI); Ref. = compared reference; * age-adjusted Charlson Comorbidity Index.

**Table 4 cancers-15-02534-t004:** Uni- and multivariate Cox analysis for progression-free and overall survival of advanced-stage III–IV, high-grade serous histology patients without tests after recurrence diagnosis.

Covariates	Patients, n	Progression-Free Survival				Overall Survival			
		Univariate HR (95% CI)	*p*-Value (Wald Test)	Multivariate HR (95% CI)	*p*-Value (Wald Test)	Univariate HR (95% CI)	*p*-Value (Wald Test)	Multivariate HR (95% CI)	*p*-Value (Wald Test)
Gene panel test result:									
No mutations	343	Ref.		Ref.		Ref.		Ref.	
BRCA 1/2 mutation	98	0.47 (0.34–0.64)	<0.001	0.47 (0.34–0.65)	<0.001	0.44 (0.28–0.67)	<0.001	0.45 (0.29–0.71)	<0.001
Other mutations	25	0.78 (0.48–1.28)	0.324	0.65 (0.40–1.07)	0.092	0.53 (0.26–1.09)	0.084	0.38 (0.18–0.78)	0.009
History of malignancy:									
No	397	Ref.		Ref.		Ref.		Ref.	
Breast cancerothers	42	0.85 (0.57–1.28)	0.436	0.80 (0.52–1.21)	0.288	1.07 (0.64–1.80)	0.797	1.26 (0.74–2.16)	0.387
	27	1.63 (1.06–2.51)	0.025	1.62 (1.05–2.51)	0.029	1.83 (1.07–3.12)	0.027	1.96 (1.15–3.36)	0.013
Surgery timing and residual disease (RD):									
Primary debulking without RD	229	Ref.		Ref.		Ref.		Ref.	
Primary debulking with RD	83	2.87 (2.10–3.91)	<0.001	2.86 (2.09–3.91)	<0.001	3.10 (2.09–4.59)	<0.001	3.17 (2.13–4.72)	<0.001
Interval debulking without RD	96	1.87 (1.38–2.54)	<0.001	1.84 (1.36–2.51)	<0.001	1.83 (1.19–2.82)	0.006	1.79 (1.16–2.75)	0.008
Interval debulking with RD	40	4.19 (2.84–6.18)	<0.001	4.55 (3.08–6.73)	<0.001	5.73 (3.57–9.21)	<0.001	6.19 (3.84–9.98)	<0.001
Diagnostic surgery	18	5.24 (3.03–9.05)	<0.001	6.06 (3.45–10.6)	<0.001	3.34 (1.65–6.77)	<0.001	3.60 (1.75–7.38)	<0.001
FIGO stage:									
III	200	Ref.		Ref.		Ref.		Ref.	
IV	266	1.38 (1.09–1.75)	0.007	1.15 (0.90–1.47)	0.247	1.34 (0.98–1.73)	0.067	1.12 (0.81–1.55)	0.484
ACCI * score:									
0–1	181	Ref.		Ref.		Ref.		Ref.	
2–3	215	0.99 (0.77–1.27)	0.950	0.95 (0.74–1.23)	0.708	1.26 (0.91–1.77)	0.166	1.20 (0.85–1.70)	0.294
>4	70	1.56 (1.12–2.17)	0.008	1.22 (0.84–1.76)	0.289	1.98 (1.29.3.05)	0.002	1.58 (0.95–2.62)	0.078

Data are presented as hazard ratio (HR) (95% CI); Ref. = compared reference; * age-adjusted Charlson Comorbidity Index.

## Data Availability

No data available.

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
