# Peer review of "Incidence and Prognostic Impact of Deleterious Germline Mutations in Primary Advanced Ovarian Carcinoma Patients"

_cancers, 2023, doi:10.3390/cancers15092534_

Round 1
Reviewer 1 Report
I would like to congratulate the Authors for the interesting study providing further evidence on the incidence and impact of of deleterious germline mutations in primary advanced ovarian carcinoma patients. Well written and designed study. However, few comments are due:
Abstract
- The authors mention that “702 patients met inclusion criteria” but they did not mention inclusion criteria
- Method used to test the genes should be mentioned
Methods
- which stage was included? If all stages were included it should be highlights. Please note that in the title authors mention only advanced stages (if all stages are included I would suggest an amendment of the title)
- Testing for all OC patients in the last 5 years or in the last 5 years of the study?
Results
- “Only 1 patient with low-grade OC histology had BRCA1/2 mutation, and further 5 patients in other genes.”: please add %
- “Mucinous OC patients showed no germline mutations (n=14).” please add %
- Please add p value in figure 1 and 2
- it would be interesting to correlate germline with somatic mutations in those patients who have somatic mutation info: can authors comment on that?
Author Response
I would like to congratulate the Authors for the interesting study providing further evidence on the incidence and impact of deleterious germline mutations in primary advanced ovarian carcinoma patients. Well written and designed study. However, few comments are due:
Abstract
- The authors mention that “702 patients met inclusion criteria” but they did not mention inclusion criteria
Response: We now added the main inclusion criteria to the abstract (page 3, sentence 58-59).
- Method used to test the genes should be mentioned
Response: Test name has been noted in the abstract (page 3, sentence 58).
Methods
- which stage was included? If all stages were included it should be highlights. Please note that in the title authors mention only advanced stages (if all stages are included I would suggest an amendment of the title)
Response: Thank you for this comment. Actually, all stages have been included in the entire cohort analyses (See FIGO stages in table 1). This Information has been now highlighted in the methods section (sentence 137, page 6). However, due to the prognosis heterogeneity of the different histology and stages, we performed a sub analysis for the latter main subgroup as noted in the results section page 11, sentences 235-236: “Due to very heterogenous survival outcome of different OC stages/histology, we conducted separate analysis for the subgroup advanced stages (stages III-IV) high grade serous histology”. We noted now this information in the methods section as well, under statistical analysis, page 7, sentence 181. Thus, we would not suggest to edit the current title.
- Testing for all OC patients in the last 5 years or in the last 5 years of the study?
Response: Testing allows to be performed for all OC patients since 2017 in Germany. We edit this sentence in page 7, paragraph 1, to make it clearer.
Results
- “Only 1 patient with low-grade OC histology had BRCA1/2 mutation, and further 5 patients in other genes.”: please add %
Response: As requested, the percentage has been added.
- “Mucinous OC patients showed no germline mutations (n=14).” please add %
Response: As requested, the percentage has been added.
- Please add p value in figure 1 and 2
Response: p-value of the figures were already noted in the results section, page 12, under survival outcome paragraph, and has been now added the figures legends.
- it would be interesting to correlate germline with somatic mutations in those patients who have somatic mutation info: can authors comment on that?
Response: Thank you for this thoughtful comment. We agree that the correlation between germline and somatic mutations for the prognosis affecting could be very interesting. However as mentioned in the discussion section, limitations paragraph (page 20): “we focused on germline mutation in the mentioned genes, given the nature of our database and missed the data on somatic mutation or homologous recombination deficiency”.
Reviewer 2 Report
The manuscript reports a lot of data from 702 patients chosen from a larger cohort. The cohort was divided in three groups A B and C (,no mutations, deleteious mutation in BRCA and mutations in genes other that BRCA). Please report this in the tables. The purpose of the study must be focused a little more in the introduction. Methods need to be more explicative expecially in the description of the statistical analysis.
Results are reported quite bad. Huge tables without legend. The authors reports many tumor characteristics without or with little description in the text. They listed the data and did statistics; it is not clear how the overall data have been analysed; SD is not reported. I presume that in the brackets the number of patiens are reported, but sometimes the age of the patients is reported. The p-value was calculated comparing what? Why SD is not reported? What is HR (hazard ratio??) in table 3? All the acronyms need to be reported and clarify. Language is not good and the manuscript is not eaasy to follow.
I recommend rejection in the present form. Authors have to reorganized all the manuscript and maybe resubmit it.
Kind regards
Author Response
The manuscript reports a lot of data from 702 patients chosen from a larger cohort. The cohort was divided in three groups A B and C (,no mutations, deleteious mutation in BRCA and mutations in genes other that BRCA). Please report this in the tables. The purpose of the study must be focused a little more in the introduction. Methods need to be more explicative expecially in the description of the statistical analysis.
Results are reported quite bad. Huge tables without legend. The authors reports many tumor characteristics without or with little description in the text. They listed the data and did statistics; it is not clear how the overall data have been analysed; SD is not reported. I presume that in the brackets the number of patiens are reported, but sometimes the age of the patients is reported. The p-value was calculated comparing what? Why SD is not reported? What is HR (hazard ratio??) in table 3? All the acronyms need to be reported and clarify. Language is not good and the manuscript is not eaasy to follow.
I recommend rejection in the present form. Authors have to reorganized all the manuscript and maybe resubmit it.
Kind regards
Response: Many thanks for the reviewer’s comments and for the opportunity to response on that. Firstly, we now noted the cohort numbers in table 1 and 2 as divided (A,B,C).
Secondly, the introduction section is quit long and includes data on ovarian cancer generally, on genetic predisposition and germline mutations for OC, on international guidelines recommendations regarding genetic testing, on the physiological plausibility of mutations and therapy effect and recently approves, on known incidence and prognostic impact of germline mutations in OC patients, and on the study aim. If the reviewer suggests to focus on other subjects, please let us know.
Thirdly, we described the main statistical tests in the methods section (page 7): “Statistical analysis was performed using GraphPad Prism (version 7) and SPSS (version 27.0, IBM Corporation, New York, USA) software. Initial analysis compared background, and tumor characteristics between the different study groups, using the chi square for categorical data or based on variable characteristics and normal distribution t-test or non-parametric Mann-Whitney test, and Anova-test or Kruskal-Wallis-H for continuous variables”. Which further information does the reviewer missing?
Fourthly, the main results have been reported in the results section, separately for the entire cohort and for the advanced stages high grade serous histology subgroup. We divided the results into patients’ characteristics and survival outcome. Each Table has an own legend (directly above the tables). The tables are obviously more explicative with more information than the text. We have chosen to show the range than the SD, if requested we can switch/add the data of SD, which has been already analyzed:
|
Characteristic |
Total 100% (n=702) |
No deleteriousmutation 76.6% (n=538) |
Deleterious BRCA 1/2mutation 17.4% (n=122) |
Other deleteriousmutation 6.0% (n=42) |
p-value |
|
Patient age, Median (range, SD) |
59 (18-87, 12.3) |
60 (18-87, 13.5) |
57 (38-85, 8.6) |
63 (30-85, 12.8) |
0.011 |
Fifthly, that’s right, in the brackets we noted the n, we edited now the description under the tables: “Data are presented as % (n) or median (range)…”. HR is hazard ratio, we rewrote now all acronyms to make it clearer. P values for Table 1 and 2 have been calculated comparing cohort A vs. cohort B vs. cohort C. For Table 3 and 4, each variable has been compared with his reference as noted in the specific table.
Finally, the whole manuscript has been edited by native English speaker.
Reviewer 3 Report
The manuscript titled 'Incidence and prognostic impact of deleterious germline mutations in primary advanced ovarian carcinoma patients', by Imterat et al. is a comprehensive and extensive study on the prevalence and clinical impact of deleterious germline mutations, in BRCA1/2 and other genes, on survival of OC patients. The authors make the interesting observation that germline mutation was detected in one-fourth of OC patients, of them one quarter observed in other genes than BRCA1/2. Germline mutations may serve as a prognostic factor and predict better prognosis for OC patients. The study is well designed to address questions with positive and negative control, the manuscript will be of great interest in the clinical oncology field. The manuscript seems to be ready for possible acceptance.
Author Response
The manuscript titled 'Incidence and prognostic impact of deleterious germline mutations in primary advanced ovarian carcinoma patients', by Imterat et al. is a comprehensive and extensive study on the prevalence and clinical impact of deleterious germline mutations, in BRCA1/2 and other genes, on survival of OC patients. The authors make the interesting observation that germline mutation was detected in one-fourth of OC patients, of them one quarter observed in other genes than BRCA1/2. Germline mutations may serve as a prognostic factor and predict better prognosis for OC patients. The study is well designed to address questions with positive and negative control, the manuscript will be of great interest in the clinical oncology field. The manuscript seems to be ready for possible acceptance.
Response: Many thanks for this positive review.
Round 2
Reviewer 2 Report
The revised version is not very different from the previous one although is a little clearer than the previous one.
There was a little improvement and could be acceptable now although the novelty is not high.